# Effective Co-Creation Tool Development for Shared Understanding and Active Communication for Patients with Head and Neck Cancer

**DOI:** 10.3390/healthcare11081180

**Published:** 2023-04-20

**Authors:** Yoo-Ri Koo, Eun-Jeong Kim, Inn-Chul Nam

**Affiliations:** 1Department of Service Design, Graduate School of Industrial Arts, Hongik University, Seoul 04066, Republic of Korea; yrkoo@hongik.ac.kr; 2Department of Industry-Academic Cooperation Foundation, The Catholic University of Korea, Seoul 06591, Republic of Korea; dodam.design.research@gmail.com; 3Department of Otorhinolaryngology-Head and Neck Surgery, Incheon St. Mary’s Hospital, The Catholic University of Korea, Incheon 21431, Republic of Korea

**Keywords:** EBCD (experience-based co-design), co-creation tool, patient experience, co-creative workshop, head and neck cancer

## Abstract

To improve the quality of life of cancer patients, patient experience (PE) must be improved along with the overall treatment process. This study aimed to develop an effective and practical co-design tool to improve the healthcare service experience of patients with head and neck cancer (HNC) in various factors. The research consisted of four phases: (1) HNC PE categories for healthcare improvement were identified through systematic review, user interviews, and observation; (2) a focus group meeting was held to materialize the card design; (3) a structured and visual card set was developed for stakeholders to share the content and discuss improvements in PE effectively; (4) to evaluate the feasibility of the developed cards, a co-creation workshop with HNC medical staffs was conducted. From the workshop using insight cards, we identified the differences in the medical staff’s and patients’ perspectives on the factors for improving HNC PE in each stage of the treatment journey. *Pat Exp Insight Cards* as experience-based co-design (EBCD) tools can be useful for stakeholders to empathize with the specific pain points and needs of patients with HNC and to discuss improvement plans efficiently.

## 1. Introduction

Patient experience (PE) is the totality of all interactions with the patient perspectives during treatment [1]. It is a significant factor in the quality of care directly related to the patient’s treatment results, safety, treatment effects, and doctor–patient interactions [2,3,4]. PE is crucial for improving healthcare because it includes everything that occurs while a patient receives treatment and covers the emotional aspect [2,5]. As a methodology for improving healthcare, co-design workshops have recently been attracting attention [2]. Co-design is a creative improvement plan for designers to improve PE with various stakeholders, such as medical staff, patients, researchers, and caregivers, including non-designers [2,6,7,8]. In particular, experience-based co-design (EBCD) designs user experiences where medical staff and patients participate together to improve healthcare services [9,10,11]. Identifying needs based on PE, seeking solutions, and presenting results are conducted through active stakeholder collaboration [12,13]. As an effective service improvement method for improving PE [12,13,14], EBCD includes patients and other stakeholders with varied expertise [15].

For stakeholders to efficiently gather and discuss improvement plans for PE, a shared and deep understanding of the collected data is essential [16]. Most of the existing PE data were collected from the perspective of the medical staff. Since it contained professional content, it took much work for service users, such as designers, patients, and caregivers, to easily access and interpret for co-design [2]. To overcome these problems, various service design methodologies have recently been used as participatory design approaches to understand and share patients’ stories [17] deeply. In addition, as a co-design methodology that can more conveniently understand PE, methods such as patient participation as co-designers [18], digital ethnography, and accelerated experience-based co-design (AEBCD) [6] are being introduced [2].

The usefulness of a card-based tool is often mentioned as a co-creation tool to efficiently derive shared understanding and creative ideation among various stakeholders [19,20]. Cards can be a source of inspiration for design ideas and a medium for conversation when stakeholders with diverse knowledge and backgrounds conduct co-design to improve PE and are helpful in the collaborative design process [21,22,23,24,25]. In addition, cards have the advantage of having relatively low production time and costs compared to other tools, and they can be used flexibly in various situations. Villalba [2] showed the advantages of cards as a co-creation tool that can arouse empathic understanding, enable the construction and sharing of knowledge, and function as a medium that promotes collaboration. It was emphasized that the card is a simple and tangible method that can be used in collaborative interactions among card users, helping stakeholders with various experts share their expertise and experiences and derive improvements in user experience.

Patients with head and neck cancer (HNC) experience difficulties in daily life, such as talking and eating [26]. Most are elderly; therefore, there are limitations in terms of direct expression and communication. They usually have deep depression and low life satisfaction owing to frequent complications and long treatment periods [27]. Therefore, improving the treatment effect of these patients and overall satisfaction with their medical experiences and quality of life is essential.

It is crucial to improve the healthcare experience of patients with HNC throughout the treatment process to increase their therapeutic effects and overall satisfaction with their medical experience and quality of life. Various stakeholders in the hospital are involved in the treatment journey of HNC, including the attending physician and other medical staff, administrative staff involved in hospitalization/discharge, and related departments involved in postoperative rehabilitation. All these factors improve the patients’ medical experiences. It is crucial to share pain points and needs felt by patients with medical service providers and seek solutions to improve the overall experience of the treatment journey. However, many realistic limitations exist to understanding the treatment and the direct participation of older patients with HNC in the co-creation process for PE improvement. Therefore, co-creation tools that indirectly represent their experiences and stakeholders with expertise can be used effectively to seek solutions throughout the patient’s treatment journey and are urgently needed.

The study aimed to develop tools to improve the healthcare experience of patients with HNC. These tools represent the experiences of those who have difficulty directly participating in the improvement process and be used in the co-creation process based on the shared understanding of stakeholders. In this study, among various tools and methods for co-creation, a card-based tool (from now on referred to as *Pat Exp Insight Cards*), which is relatively easy to produce and has the potential to be used flexibly in various situations, was developed. 

## 2. Materials and Methods

This study was conducted in four stages. The first phase entailed structuring the contents of the card set. We derived the HNC PE factors and specified the patient’s needs for each factor through key insights and quotations. A general PE factor list was derived through a systematic review, interviews and observations conducted with patients with HNC and their caregivers, and insights and citations were analyzed for each PE factor. The systematic review complied with the PRISMA guidelines. We searched for papers published between 2015 and 2021 in four electronic databases: Google Scholar, PubMed, Web of Science, and Taylor & Francis. Search keywords used ‘healthcare’, ‘service design’, ‘patient experience factor’, ‘quality improvement’, and ‘patient-centered care’, and the search range was expanded by including similar words. Further, 7008 papers were searched in the first round, and 6185 papers were selected, excluding the papers for which abstract and author could not be found, as well as the number of overlapping papers. Papers that directly or indirectly analyzed factors that affect PE improvement were included in the inclusion criteria. The exclusion criteria included papers in which PE factors were presented as methods or criteria for specific prototype development or evaluation of patient satisfaction. Further, 135 papers were selected through the abstract review using these criteria, and 23 papers were finally selected through a full-text review. From the selected papers, we identified the general PE factor list.

Semi-structured interviews and observations of the consultation were conducted with 31 participants (including patients, caregivers, and medical staff) to reflect on the specific needs of patients with HNC for each identified PE factor. The interview questions consisted of 25 items, including patient condition, treatment journey experience, route of visits within the hospital, time spent for treatment, contents of the information received during the treatment process, communication difficulty, psychological state of the patient and caregiver, the physical environment of the hospital, feedback on the PE or satisfaction, and experience of patient education. For observation, if the consent of the patient and caregiver was obtained, the researchers participated in the consultation and observed the entire treatment process. Researchers observed and filmed the clinical environment, tools and objects, interactions between doctors and patients, and recorded conversations.

Ethical approval was obtained before we conducted the study. An otolaryngologist completed the recruitment of patients and caregivers who participated in the study. When a patient with HNC visited the hospital, the purpose of the study and the contents of the user research was explained. When the patient and his/her caregiver expressed their intention to participate, the contact information was delivered to the researchers to arrange an interview and observation schedule. All data were collected with the consent of patients and caregivers, and key insights and quotes were carefully selected from the full transcripts of the interview data.

The second phase was the design ideation stage for card development, and key considerations and design directions were set to develop the *Pat Exp Insight Cards* for patients with HNC. Through a two-hour focus group meeting (offline, 28 January 2022) with two otolaryngologists, one otorhinolaryngology researcher, one service design professional, and three service design researchers, the compatibility of the factor list and hierarchy of items were reviewed, and the design development direction was discussed. The meetings were divided into two sessions of 60 min each. In the first session, the items in the list of PE factors were appropriately configured to structure the card. The second session focused on whether the composition of the card content was appropriate and whether the expression of the card design effectively delivered the content. To efficiently conduct the meeting, a discussion was conducted by specifying the following questions for setting the design direction (DD) based on the four design goals (DGs) presented in Table 1.

DD 1. How should the patient-centered perspective and needs be reflected in the card for stakeholders to understand and empathize with the experience of patients with HNC? (DG 1)

DD 2. What is an effective information delivery technique that helps stakeholders understand and discuss PE factors? (DG2)

DD 3. What expression technique helps participants understand the content of the card intuitively? (DG 3)

DD 4. How should the various viewpoints and opinions of stakeholders participating in decision-making for each PE factor be expressed? (DG4)

An initial card prototype was developed based on the meeting results. To refer to the card design development process, Villalba’s case study [2] of designing a co-creation card tool was reviewed. She developed an 85-card set called *Healthcare Experience Insight Cards- ‘living with Diabetes Edition’* as a co-creation tool for diabetic patients. The card set comprised four items: theme, detailed insight, quote, and reference. The topmost theme contained topics related to PE identified through literature reviews. Under this theme, insights for improvement by the PE category were introduced, and, below that, quotes from patients extracted from the literature were presented along with references (card link: https://link.springer.com/article/10.1007/s40271-018-0315-7 (accessed on 15 February 2022)).

The third phase was the card visualization stage, in which the previously developed prototype was refined in detail. Doctors’ feedback was received twice on the modified prototype, and the overall composition and contents of the card items were reviewed, including the appropriateness of insights and quotes for each factor and details for efficient design delivery.

In phase four, a usability test was conducted using the completed card prototype, and the effectiveness of the card was verified. For evaluation, a co-creation workshop (25 March 2022) with the medical staff was conducted for two hours. The factors to be considered for improving HNC PE were divided into the patients’ and medical staff’s perspectives while discussing the importance, priority, and related cases. To maintain consistency of the stakeholders’ viewpoints, the participants who attended the second-phase meeting were involved in the workshop. A total of 35 EBCD cards were introduced to the medical staff, and each factor card considered important from the perspectives of the medical staff and patient was selected. When selecting a card, the relevant factor cards were pinned according to the stages of the HNC treatment journey. If the same card was repeatedly selected for each stage of the journey, extra sets of cards were provided when necessary. The priority and importance from the perspective of the medical staff were marked on the card, and they had time to exchange opinions about the reason for their selection. After selecting the factors that should be considered necessary for each stage of the treatment journey from the perspective of the medical staff, factors considered necessary from the patients’ perspective were discussed separately. Instead of having the patients directly participate in the workshop, medical staff who interacted closely with patients selected key factors for each stage of the journey based on their medical experience. To reflect the patient’s view, the medical staff selected the factors based on the complaints or requests that patients frequently express to the medical staff during treatment. By distinguishing this from their point of view and placing the cards on the journey map, the difference between the medical staff’s perspective and the patient’s perspective were compared.

After the workshop, feedback on the usefulness and value of the card was received from the medical staff, and further improvements to the design were discussed. Based on the results discussed, the final *Pat Exp Insight Cards* for patients with HNC, consisting of 35 items, were developed.

The flow chart of the research is shown in Figure 1.

## 3. Results

### 3.1. Phase 1: Structuring Card Contents

Before deriving the HNC PE factors, a list of general PE factors for QI improvement was identified through a systematic review. By examining the PE factors mentioned in the twenty-three selected papers, six categories, eighteen subcategories, and eighty-one specific PE factors were identified.

To select items applicable to patients with HNC and to reflect their needs in detail among the 81 identified items, interviews and observations were conducted targeting patients with HNC and their caregivers. From the user data, we summarized the needs of patients with HNC for each PE factor into key insights through qualitative content analysis (HNC includes various subsites. However, there is no difference between subsites in the treatment principles, the process patients experience after treatment, and problems and complications. Therefore, since the patient’s needs are not significantly different for each subsite, the study was conducted without classifying by subsite.). As a result, except for items where the needs of patients with HNC were not directly mentioned, a total of six categories (*Practice, Physical needs, Psychological needs, Social needs, Practical needs, and Information needs*), sixteen sub-categories, and forty-three PE factors were found to have a direct relationship with the needs of patients with HNC (Phase 1 in Table 2).

### 3.2. Phase 2: Card Development: Design Ideation

#### 3.2.1. Set Up Key Considerations and Design Direction

Key considerations and design directions were set to develop a card set based on the previously identified 43 HNC PE factor lists. by referring to the contents of *Healthcare Experience Insight Cards* developed by Villalba [2]. The key considerations were set as shown in Table 1 to develop an effective design for representing the experience of patients with HNC and discussing effective improvement plans through stakeholders’ shared understanding and effective brainstorming.

#### 3.2.2. Focus Group Meeting for Card Development

A focus group meeting was held to discuss the appropriateness of the content composition and design direction of the *Pat Exp Insight Cards* for patients with HNC. 

As a result of reviewing the contents of the PE factor list, the initial categories were reclassified by filtering redundancy between items and integrating items with similarity in meaning. As a result, a new category of *Interaction* was added to the existing six categories, and the names of the sub-categories and PE factors were partially changed to convey a precise meaning. According to the reclassification, seven categories, sixteen sub-categories, and thirty-six PE factors were systematized (Phase 2 in Table 2). For example, ‘Hard to understand,’ ‘Easy to understand,’ and ‘Interaction between doctor and patient’ of ‘Knowledge’ highlight the aspects that the patient does not understand the explanation of the medical staff well. Thus, we integrated the three items into ‘hard to understand’ to convey precise meaning to the subject. In ‘Information’, ‘What patients want to know,’ ‘What else to know,’ and ‘Alert effects of treatment’ were integrated into ‘What patients want to know’ as they corresponded to the information the patient wants or needs to know. ‘Understanding patient’s life situation’ and ‘Building a trusting relationship’ under ‘Communication’ and ‘Breaking the ice’ under ‘Respect’ and ‘Cure vs. Relief’ under ‘Respect’ were judged to be the content corresponding to the respectful attitude of the medical staff to relieve the patient’s anxiety and build a positive relationship, so these items were moved into a new category of ‘Interaction’.

The discussion results of the four design directions of the card are as follows: First, to reflect patient-centered perspectives and needs, it was necessary to present specific pain points and opinions for each PE factor. It was agreed that extracting cases that best explained each PE factor from the patient’s interview and observation data and then presenting a quotation would be effectively convey the patient’s voice.

Second, it was discussed that an effective delivery method for best understanding PE factors was presenting categories and sub-categories together instead of showing one of them. This facilitated a better understanding of the classification system and a higher concept level for each factor. This method was chosen instead of introducing only the PE factors on the card. However, it was agreed that it was necessary to visually show the hierarchy to distinguish the relationship between each factor clearly.

Third, to facilitate an intuitive understanding of the card’s contents, it was suggested to implicitly express specific situations using illustrative images instead of realistic images, such as photographs, to make it more effective. However, the meeting participants recognized that interpreting the core meaning was crucial to clearly and symbolically convey the concept of each PE factor when creating an image. It was emphasized that the card design image should be an action-based emotional exchange and that interactions could be read well in the image.

Fourth, to express the diverse perspectives of stakeholders, it was preferred that various direct and indirect stakeholders be included. Although the direct subject-centered type is suitable for patients to focus on, it was judged to be helpful to include several direct and indirect stakeholders for discussion. However, to simplify the format, the stakeholders should be classified into three types: medical staff, patients, and caregivers, where each type was presented as an icon to enhance visual distinguishment. For example, medical staff icons were developed for HNC doctors and medical staff in other departments. The insights suggested for each card were presented in one summarized sentence to facilitate an intensive discussion. It was agreed that the most common cases were presented instead of exceptional cases to consider general-purpose usage.

### 3.3. Phase 3: Card Development: Visualization

#### 3.3.1. Refinement of *Pat Exp Insight Cards* Based on the Feedback from Doctors

The HNC specialists carefully reviewed the list of 36 PE factors modified through the focus group meeting. They reviewed the appropriateness of the insights and quotes developed for each factor and provided feedback on the factors’ structure and the hierarchy’s suitability.

As shown in Table 2, No. 3 merged the ‘disconnected care system’ and ‘after discharge’ of phase 1 into one. In No. 5, the term ‘issues with physical symptoms’ was modified to ‘issues related to complications’ considering that HNC patients had frequent complications. In No. 19–22, among the items belonging to social needs in phase 1, items related to the relationship between doctors and patients and doctors’ attitudes towards patients were separated from the category of social needs and moved into the newly established interaction category. Items 27 and 28 originally belonged to practical needs but were moved to the information needs category as they corresponded to the contents of delivering treatment information to patients. In No. 31, the expression was modified by integrating various information that the patient wanted to know into ‘comprehensive treatment information’. Finally, the expression used in the previous phase was modified more concisely for No. 32 and summarized as ‘information about care plans’.

As a result, quotations were replaced with general cases instead of exceptional cases and modified. Names of PE factors were changed to convey precise meaning, integration, and changes between similar items, and insight content was modified. Based on this feedback, the list was revised into seven categories, fifteen sub-categories, and thirty-five PE factors (Phase 3 in Table 2).

First, the category of *Practice (1)* includes perspectives of both patients and medical staffs. It deals with factors related to the hospital’s administrative and procedural systems, including three sub-categories: System (1-1) aspects, including complex and disconnected systems (patients are confused because each hospital has different systems) and clinician’s workload (the same explanation has to be repeated every time the caregiver changes), Coordination (1-2) regarding the disconnected care system (difficulty in hearing conflicting opinions between departments or obtaining comprehensive information), and Skill (1-3) related to the diagnostic procedure and guidance (preliminary evaluation must be conducted for accurate cancer diagnosis).

Second, the *Physical needs (2)* category consisted of items about the physical pain felt by the patient, including two sub-categories: Physical support (2-1) regarding issues related to complications (wanting side effects after surgery to be minimized) and Physical symptoms (2-2) that cover both in case of emergency (possibility of emergency, such as shortness of breath, and guidance on how to respond) and the contents of physical symptoms and pain after surgery (discomfort to breathe, eat, and talk after surgery).

Third, the *Psychological needs (3)* category deals with the patient’s emotional aspects, introducing the negative psychology of patients during the treatment process and the encouragement and support of medical staff. It includes two sub-categories: Emotional support (3-1), including the doctor’s empathy (the doctor shows a positive attitude toward patients) and motivates the patient (explaining that his/her condition can improve after surgery through rehabilitation), and Psychological symptoms (3-2) dealing with feelings of frustration (extreme psychological pain from multiple surgeries and fear of permanent side effects) and the emotional impact of results (willingness to recover or increased depression depending on the outcome of the surgery).

Fourth, the *Social needs (4)* category consists of items related to interactions between patients, caregivers, and medical staff and includes two realistic aspects, such as economic support and rehabilitation guide related to treatment: Communication (4-1) consisting of too old to understand (older patients have difficulty understanding explanations and are passive in expressing their opinions), disagreement between family members (the timing may be delayed due to differences in opinion between the patient and the caregiver on the treatment method), and tools to communicate (non-verbal communication tools are needed due to difficulties in vocalization and articulation after surgery), and Support & Involvement (4-2) dealing with maintaining daily life (requires a guide on exercises and habits that patients must manage on their own after discharge), motivated by clinicians (the encouragement and consolation of medical staff after surgery increases the will to treat), having family at the bedside (wanting comfort and encouragement from close people), and finance & insurance (high burden of non-coverage costs and long-term treatment costs).

Fifth, the *Interaction (5)* category includes personal and emotional aspects rather than social needs to build trust and bond between patients and medical staff covering the sub-category of Respect (5-1): understanding the patient’s life situation (requires an understanding of the patient’s circumstances, such as where they live and work), building trusting relationships (diagnosis based on clear evidence, encouragement by medical staff, communication channels through which patients can express their opinions and help build mutual trust), breaking the ice (it is necessary to reduce the patient’s tension and build a bond with small talk), and clinicians’ positive attitude (it is essential to deliver objective facts with a positive attitude rather than showing excessive hope).

Sixth, the *Practical needs (6)* category deals with items related to the convenience of the patient’s treatment process: Access to care (6-1) covering the minimum waiting time (patient wants to be treated as quickly as possible), ASAP (wanting to advance the surgery schedule as quickly as possible), useful helper (need help with guidance and appointments for new patients), and easy to access (prefer a hospital close to home or convenient transportation).

Finally, the *Information needs (7)* category includes four subcategories: Access to information (7-1), meaning the more information, the better (the patient tries to find as much information as possible on the internet), and keeping patients informed (requires periodic guidance on treatment schedules, etc.), Knowledge (7-2) related to information overload and difficulty in understanding (challenging to understand if you receive a great deal of information at once), Information content & extent (7-3) providing up-to-date information (you want prompt and continuous guidance on treatment progress), comprehensive treatment information (wants to know comprehensive information about the entire treatment process), information about care plans (you need a guardian right after surgery and you need to take care of yourself after discharge), limited information on the internet (challenging to find accurate and necessary information on the internet), Education (7-4), including patient education for prevention or rehabilitation (requires long-term rehabilitation training and education for forming preventive habits), and patient education for self-care (education that can systematically help with long-term rehabilitation training that is difficult to carry out alone after discharge is needed).

#### 3.3.2. Card Visualization

The card was designed based on the final list of *Pat Exp Insight Cards* for patients with HNC. The design principles were applied based on the previously established design direction and results of the focus group meeting. The card layout comprised categories, sub-categories, PE factors, quotations, insights, illustrated images, stakeholder icons, and code/card numbers.

The title of each category is shown at the top of the card front, indicating that it corresponds to the highest level in the HNC PE category. Color codes were applied to each category to make it easier for users to identify significant topics. The front of the card was designed to highlight the title to help users understand the concept and classification system of the subject. The title of the sub-category is placed below the category title, and the PE factor and related image are placed in the center to highlight the subject. The notation was written in Korean and English to consider the usability and convenience of users.

Insights and quotations for each PE factor are placed on the back of the card for users to understand the patients’ needs in detail and access their actual voices. Each insight and quote reflects the perspectives of medical staff, patients, and caregivers in various ways, and each stakeholder is visualized as an icon to distinguish between them. For each card, the card number is marked on the lower right corner of the back page, and the code number is marked on the lower left corner of the front page. Therefore, even if the cards are mixed, the types can be distinguished and classified using numbers (Figure 2).

The 35 *Pat Exp Insight Cards* for patients with HNC developed through the above process introduce the items in the order of category, sub-category, and PE factor on the front of the card to clearly understand the criteria of the subject to be discussed. The PE factor is emphasized in large and bold letters, ensuring the subject stands out well for each card. An illustration image visually symbolizing the subject was developed for each factor, and the code number according to the category classification system was marked. On the back of the card, insights into improving PE based on the needs of patients with HNC were introduced. Quotes were added to convey the stakeholders’ opinions related to the insights. For convenience, the number of cards was written at the bottom. The complete *Pat Exp Insight Cards* list can be found in Appendix A.

### 3.4. Phase 4: Usability Test

#### 3.4.1. Co-Creation Workshop Using the Pat Exp Insight Cards

A co-creation workshop was conducted with the HNC medical staff to evaluate the usefulness and value of the *Pat Exp Insight Cards* for patients with HNC (Figure 3). It was conducted to identify specific factors to improve the medical experience of patients with HNC and to discuss the relative importance of each stage of the patient’s treatment journey and the stakeholders’ perspectives. 

As a result of the workshop, the medical staff selected different factors that were considered important for each stage of the journey (Figure 4). In addition, since each medical team has a different field of responsibility or a different stage of communication with patients, it was possible to have a detailed discussion focusing on the period in which one is in charge. Table 3 compares the differences in PE factors that are considered important from the perspectives of doctors and patients at each stage of the treatment journey for HNC. For example, in the preoperative counseling stage, the patient discusses and decides on the direction of treatment after the cancer diagnosis with the doctor. At this time, the most important factor to consider from the perspective of medical staff was *Building a trusting relationship (Card No. 20).* Regarding this factor, the doctor pointed out that it is difficult to have an interactive discussion with the patient if they show an aggressive and distrustful attitude. It was also mentioned that, when a patient trusts and relies on a doctor, there is a strong tendency for decisions regarding treatment direction and future treatment processes to be made continuously.

On the other hand, the factors for improvement from the patient’s perspective that medical staff understand based on medical experience were identified as the *Doctor’s empathy (No. 8), Emotional impacts of results (No. 11), Motivating the patient (No. 9), and Feeling frustrated (No. 10).* These factors were selected because patients frequently show very negative and hopeless feelings toward the medical staff. Patients generally want their doctors to provide them only hopeful stories; however, from the perspective of doctors, who must deliver information based on objective facts, the difficulties of delivering information positively are also discussed.

In short, as a result of the workshop using the *Pat Exp Insight Cards*, it was confirmed that the difference in perspective between medical staff and patients was clearly revealed. While the patient had a strong desire to be emotionally supported by the doctor, the medical staff appeared to emphasize forming a bond with patients and improving their understanding to obtain efficient and successful treatment results. In addition, by visually and structurally understanding the differences in viewpoints among stakeholders through cards, it was possible to empathize and compromise with each other’s opinions more efficiently.

#### 3.4.2. Evaluating the Value of *Pat Exp Insight Cards* for a Co-Creation Tool

Participants evaluated the effectiveness and value of the cards in three main aspects at the end of the workshop.

First, the voices of patients with HNC could be heard in detail through quotations presented on the cards. From the perspective of medical staff, who usually have difficulty obtaining direct feedback from patients other than complaints about treatment results, it was suggested to be very useful and exciting because the patient’s point of view could be accessed through various factors. In addition, conducting user research to improve PE requires considerable time, effort, and money. By transferring this research data into card content, the advantage of obtaining efficiency is that various participants can quickly and conveniently understand and share the patient’s needs. It was also found that the medical staff could participate in improvement discussions without feeling reluctant. They could indirectly meet the requirements without receiving direct patient complaints or attacks. In addition, it was evaluated as a positive aspect that practical discussions are possible during the COVID-19 pandemic, where direct contact with older patients is a concern.

Second, participants involved in improving PE were generally confined to their specific areas of responsibility to meet and address patient needs. There are few opportunities to think about and approach the overall treatment journey and various PE factors in an integrated manner. The *Pat Exp Insight Cards* comprise seven categories that include various aspects, such as the hospital system, patient’s physical/psychological needs, doctor–patient interaction, and patient education and support. This allowed workshop participants to examine various factors other than the fields they knew comprehensively. For example, the medical staff who participated in the workshop had different points of contact (touch points) to meet and communicate with patients, leaving them no opportunity to think about anything other than the voices and requirements of patients, limited to their stage of the journey. Through this co-creation workshop, by meeting and understanding the needs of patients in fields where they are not in charge, the card was highly valued by the participants as it provided an opportunity for flexible communication between the medical staff. Additionally, by identifying the importance and priority of each factor based on the actual pain point of the patient, an initiative was provided to discuss and derive ideas on the aspects that should be improved first.

Third, because shared understanding is based on the specific pain point of the patient in the process of discussing the improvement of PE by gathering experts from various fields, it did not end with the development of abstract ideas but confirmed the possibility of presenting tangible solutions at a concrete and realistic level. For example, by sharing pain points, response know-how, and realistic PE situations accumulated by medical staff with other participants, experts in other fields can enhance their understanding and knowledge of healthcare services. They can also look at problems from a new perspective and explore ideas from their perspective. Medical staff can provide realistic advice on the various potential solutions explored in this process, thereby developing feasible outcomes.

Based on the above three aspects, the effectiveness and value of the *Pat Exp Insight Cards* were confirmed. Workshop participants evaluated the card set developed for shared understanding and effective brainstorming in different fields as a meaningful tool for value co-creation between patients and doctors and creative interdisciplinary collaboration of various interests. In particular, in a realistic situation where collaboration between departments and systems of complex and specialized hospitals is difficult, medical staff emphasize the effectiveness of the card set in terms of providing a co-creation tool for active collaboration and presenting an integrated perspective in which medical staff from various fields can meet and actively discuss various factors of PE.

## 4. Discussion

In this study, an effective EBCD card (*Pat Exp Insight Cards*) was developed to improve patients’ healthcare experiences with HNC. As a result, the value and effect of cards as a co-creation tool for improving patient-centered healthcare experiences could be confirmed as follows.

We identified primary PE factors through a literature review to provide various factors for improving HNC PE. Based on user research data targeting patients and their caregivers, we identified seven categories, fifteen sub-categories, and thirty-five PE factors. The identified categories of PE factors were developed as a graphic card deck so that users with various knowledge and expertise could participate and brainstorm effectively to improve PE.

First, this study materialized PE factors derived from a review of previous studies by linking them to actual patients’ needs. The voices and needs of patients with HNC who have difficulty directly participating in improving PE were indirectly conveyed. We actively reflected on the data collected through empirical research, such as interviews and observations, in the cards’ contents and presented them as primary data to understand the experience factors. Through this, members who participated in the co-design to improve PE were able to identify the specific and practical needs of patients with HNC based on various factors.

Second, the primary user data for improving PE were categorized by the PE factor and transferred into cards to be continuously used in improving QI rather than being a one-off. Generally, user research to improve the QI requires time, money, and effort. Thus, recording and cataloging the collected user data and converting them into structured visual materials, such as cards, can lead to significant economic savings.

Third, experts in various fields can gather and actively and creatively solve problems using visual images and easy-to-understand terms, knowledge, and information in the specialized and complex medical field, making it easy for all participants to access and share knowledge. When forming a multidisciplinary team to improve QI, finding a starting point for communication is difficult because effective brainstorming based on a shared understanding can be challenging. The EBCD card can mediate in flexible and efficient collaboration in the medical field where there is a conflict of interest between specialized fields. As such, the value of the card developed in this study as a co-creation tool is supported by the claim that the card is helpful as a co-creation tool among the multidisciplinary team for brainstorming PE improvement mentioned by many researchers [2,19,20,21,22,23,24,25]. In particular, compared to Villalba’s [2] development of an insight card for diabetic patients, this study has distinction and strength because it targeted patients with HNC who have a low quality of life and generally low satisfaction with treatment due to long-term treatment period, frequent complications, and difficulty in daily behavior.

Fourth, continuously accumulating user survey data of patients with HNC is vital for long-term data management and use. Since the PE factor items presented in the EBCD card can act as a standard for collecting, accumulating, and classifying user voices and needs, it can serve as a framework or platform for building a PE database.

A limitation of this study is that the collection of user data to improve the experiences of patients with HNC is in the early stages. Thus, continuous data collection and accumulation should be conducted by targeting more patients, caregivers, and stakeholders. In addition, considering the flexible and practical use of EBCD cards, tests, and evaluations under varied conditions according to the participation of various stakeholders and the purpose of improvement should be conducted.

## 5. Conclusions

In this study, the *Pat Exp Insight Cards* were introduced as a co-creation toolkit that indirectly represents the patients’ opinions with HNC and allows various stakeholders to collaborate effectively based on shared understanding. In the medical field, where collaboration is mutual and challenging, communication is limited owing to conflicting interests in complex and specialized fields. Efficient sharing of knowledge and opinions to improve PE with other stakeholders positively impacts patients’ quality of services and positive treatment outcomes. The systematic collection and accumulation of PE data are difficult to access; hence, flexible sharing and discussion of data knowledge among stakeholders are essential. Therefore, efficient tools such as the EBCD card developed in this study are valuable to improve the quality of PE. Based on the results of this study, *Pat Exp Insight Cards* is expected to be used as an active co-creation tool for other cancer patients in addition to patients with HNC or for improving the healthcare field.

## Figures and Tables

**Figure 1 healthcare-11-01180-f001:**
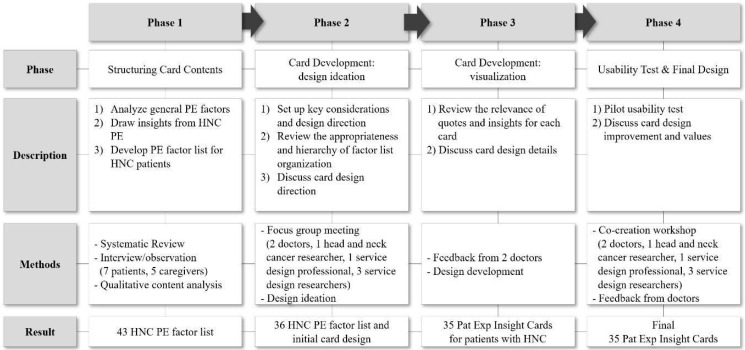
Flow diagram of study.

**Figure 2 healthcare-11-01180-f002:**
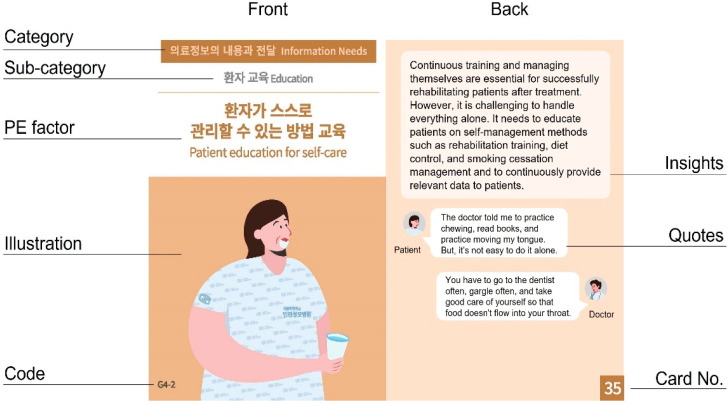
Design layout of *Pat Exp Insight Cards* for patients with HNC.

**Figure 3 healthcare-11-01180-f003:**
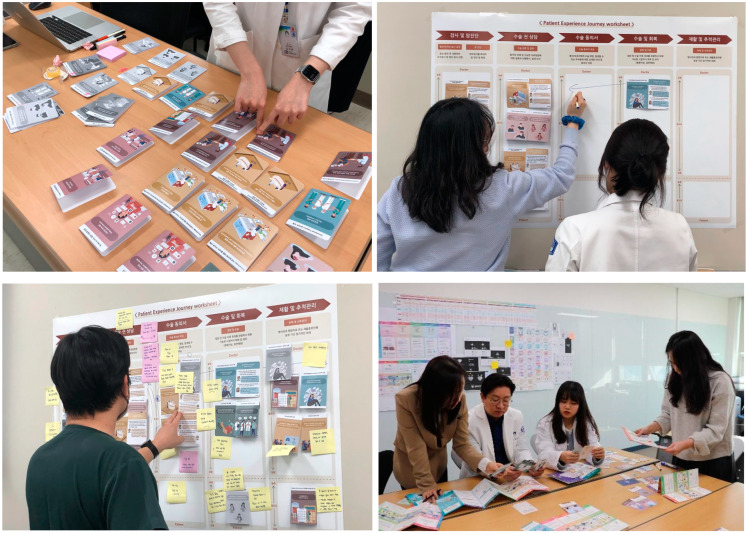
Co-creation workshop to improve HNC PE.

**Figure 4 healthcare-11-01180-f004:**
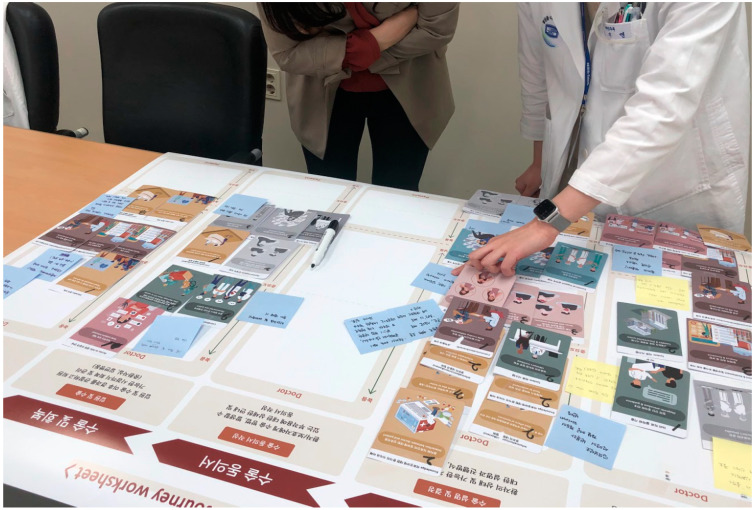
PE factors allotted on the HNC patient journey map showing the importance level.

**Table 1 healthcare-11-01180-t001:** Key considerations and design direction for ‘*Pat Exp Insight Cards* for patients with HNC’.

Code	Design Goal (DG)	Key Consideration	Design Direction
DG1	HNC PE representation from the patient-centered perspective	Materialize the needs of patients with HNC	Presenting the needs of patients with HNC by PE factor as specific insights
DG 2	Shared understanding of PE factors among stakeholders	Structuring Experience Factors	Structured visualization of card contents considering the hierarchy and attributes of PE factors
DG 3	Effective brainstorming and visual thinking	Forming empathy based on cases of HNC PE	Providing relevant images that explain the topic well and quoting the stakeholder’s narrative directly
DG4	Co-creative interdisciplinary collaboration	Encouraging active participation of various stakeholders	Introducing insights and quotes by PE factor from various perspectives by stakeholders

**Table 2 healthcare-11-01180-t002:** The list of *Pat Exp Insight Cards* for patients with HNC developed by phases *.

Phase 1			Phase 2			Phase 3			
Category	Sub-Category	PE Factors	Category	Sub-Category	PE Factors	Category	Sub-Category	No.	PE Factors
*Practice*	System	Complex & disconnected system	*Practice*	System	Complex & disconnected system	*(1)* *Practice*	(1-1) System	1	Complex & disconnected system
	Coordination	Clinician’s workload			Clinician’s workload			2	Clinician’s workload
	Care Plan	Disconnected care system		Coordination	Disconnected care system		(1-2) Coordination	3 *	Disconnected care system
		After discharge (from ICU to wards)		Care plan	Disengaged medical staff				
	Skill	Diagnostic procedure & guidance		Skill	Diagnostic procedure & guidance		(1-3) Skill	4	Diagnostic procedure & guidance
*Physical needs*	Physical support	Issues with physical symptoms	*Physical needs*	Physical support	Issues with physical symptoms	*(2)* *Physical needs*	(2-1) Physical support	5 *	Issues related to complications
		In case of emergency		Physical symptoms	In case of emergency		(2-2) Physical symptoms	6	In case of emergency
	Physical symptoms	Physical symptoms & pain after surgery			Physical symptoms & pain after surgery			7	Physical symptoms & pain after surgery
*Psychological needs*	Emotional support	Doctor’s empathy	*Psychological needs*	Emotional support	Doctor’s empathy	*(3)* *Psychological needs*	(3-1) Emotional support	8	Doctor’s empathy
		Motivate the patient			Motivate the patient			9	Motivate the patient
	Psychological symptoms	Feeling frustrated		Psychological symptoms	Feeling frustrated		(3-2) Psychological symptoms	10	Feeling frustrated
		Emotional impact of results			Emotional impact of results			11	Emotional impacts of results
*Social needs*	Communication	Too old to understand	*Social needs*	Communication	Too old to understand	*(4)* *Social needs*	(4-1) Communication	12	Too old to understand
		Disagreement between family members			Disagreement between family members			13	Disagreement between family members
		Tools to communicate			Tools to communicate			14	Tools to communicate
	Support & involvement	Maintaining daily life		Support & Involvement	Maintaining daily life		(4-2) Support & Involvement	15	Maintain daily life
		Motivated by clinicians			Motivated by clinicians			16	Motivated by clinicians
		Having family at the bedside			Having family at the bedside			17	Having family at the bedside
		Finance & insurance			Finance & insurance			18	Finance & insurance
	Communication	Understanding patient’s life situation	*Interaction*	Respect	Understanding patients’ life situation	*(5)* *Interaction*	(5-1) Respect	19 *	Understanding patients’ life situation
		Building a trusting relationship			Building a trusting relationship			20 *	Building a trusting relationship
	Respect	Breaking the ice			Breaking the ice			21 *	Breaking the ice
		Cure vs. relief			Clinicians’ positive attitude			22 *	Clinicians’ positive attitude
*Practical needs*	Access to care	Minimum waiting time	*Practical needs*	Access to care	Minimum waiting time	*(6)* *Practical needs*	(6-1) Access to care	23	Minimum waiting time
		ASAP			ASAP			24	ASAP
		Useful helper			Useful helper			25	Useful helper
		Easy to access			Easy to access			26	Easy to access
	Access to info	The more(information), the better	*Information needs*	Access to info	The more(information), the better	*(7)* *Information needs*	(7-1) Access to info	27 *	The more (information), the better
		Keep patient informed			Keep patient informed			28 *	Keeping patients informed
*Information needs*	Knowledge	Hard to understand; Easy to understand		Knowledge	Information overload & hard to understand		(7-2) Knowledge	29	Information overload & difficulty in understanding
		Interaction between doctor and patient							
	Information	Information overload							
		Provide up-to-date information		Information	Provide up-to-date information		(7-3) Information (content & extent)	30	Provide up-to-date info
	Knowledge	Not expect to have cancer			What patients want to know			31 *	Comprehensive treatment information
	Information	What patients want to know; What else to know; Alert effects of treatment							
	Knowledge	Conflicting interpretations (of treatments)		Education	Care information after discharge (what to do after discharge)			32 *	Information about care plans
	Information	Care information after discharge (what to do after discharge)							
		Limited information on the internet		Information	Limited information on the internet			33	Limited information on the internet
	Education	Patient education for prevention or rehabilitation		Education	Patient education for prevention or rehabilitation		(7-4) Education	34	Patient education for prevention or rehabilitation
		Patient education for self-care			Patient education for self-care			35	Patient education for self-care

Items marked with * indicate the changed PE factor compared to the previous phases.

**Table 3 healthcare-11-01180-t003:** The sample result of the workshop showing the importance of PE factors along the patient journey *.

Treatment Phase	Stage 1: Studies for Cancer Diagnosis and Staging	Stage 2: Pre-Operative Counseling	Stage 3: Getting Operation Consent	Stage 4: Surgery and Recovery	Stage 5: Rehabilitation And Follow-Up
	Recognizing Symptom and Being Diagnosed with Cancer	Explaining and Discussing Treatment	Explaining the Surgery Process and Methods and Signing a Consent	Hospitalizing and Recovering after Surgery	Undergoing Rehabilitation and Visiting the Hospital for Follow-Up
Perspective of medical staff	***Doctor’s empathy (8)*** (When a cancer diagnosis is made to a patient, the patient receives a great shock, so psychological stability should be given priority before proceeding with treatment.)	***Building a trusting relationship (20)*** (Medical staff should not only deliver objective facts but also keep a positive attitude based on empathy and trust.)	***Emotional impacts of results (11)*** (After deciding on surgery, the patient is anxious because it is difficult to predict the outcome.)	***Keeping patients informed (28)*** (Information about the treatment results, progress, and condition of the patient should be timely provided.)	***Patient education for self-care (35)*** (A platform should be prepared to share related information by operating a patient association, including medical staff.)
	***Diagnostic procedure & guidance (4)*** (Since various tests are required for an accurate cancer diagnosis, it is necessary to explain well so that the tests can be conducted without omission.)			***Provide up-to-date info (30)*** (Patients and caregivers want to be informed of the progress of treatment quickly and immediately.)	***Patient education for prevention or rehabilitation (34)*** (It is important to increase the patient’s will, and smoking cessation and sobriety management in daily life are necessary.)
	***Too old to understand (12)*** (It is difficult for older patients to understand explanations, and, since repeated explanations must be provided to caregivers, an easy way to communicate is needed.)				
Perspective of patient	***The more, the better (27)*** (Patients want as much information as possible to make the best decision)	***Doctor’s empathy (8)*** (For the patient to trust the doctor, it is important to build a relationship through the doctor’s empathy)	***Emotional impacts of results (11)*** (The patient develops anxieties that he or she may die during the surgery.)	***Physical symptoms & pain after surgery (7)*** (Patients prioritize relieving pain right away, and doctors consider the biopsy results more important because postoperative pain can be controlled with medication.)	***Maintain daily life (15)*** (Patients feel a great burden about managing themselves after being discharged and want the treatment they received at the hospital to be extended at home as well.)
	***ASAP (24)*** (The patient wants to advance the date of surgery as much as possible.)	***Emotional impacts of results (11)*** (Patients are anxious about unpredictable treatment outcomes.)			
	***Minimum waiting time (23)*** (Patients want to receive treatment on time, if possible, and reduce unnecessary waiting time.)	***Motivate the patient (9)*** (Patients should be motivated to actively engage in treatment.)			
		***Feeling frustrated (10)*** (Patients are greatly shocked after a cancer diagnosis and do not know what to do.)			

* This table introduces some of the PE factors that are considered important by medical staff and patients for each stage of the journey of a patient with HNC and their reasons as examples. The text in italics corresponds to the number and item of the PE factor card selected as important at that stage, and the explanation in parentheses summarizes the reason for selecting the item.

## Data Availability

The data that support the findings of this study are available on request from the corresponding author. The data are not publicly available due to privacy or ethical restrictions.

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
