# Peer review of "Effective Co-Creation Tool Development for Shared Understanding and Active Communication for Patients with Head and Neck Cancer"

_healthcare, 2023, doi:10.3390/healthcare11081180_

Round 1
Reviewer 1 Report
Introduction: Page 2, lines 50-65 appear to be methods rather than an introduction. Could you add citations and literature on page 2, lines 69-74? The research problem needs to be clarified with evidence and citations. Page 2-3, lines 94 to 102 regarding research questions, I don't think this is necessary since you have already stated the purpose of the study.Methods:
As stated on page 3, line 111, "... four electronic databases," which databases? , what terms were used? You don't need to mention the databases in your results (line 148).
Results: Page 5, lines 166-175, these are not your results, so please focus on your results. I would move that to the methods section. Instead of mentioning the numbers on page 6, please add the RQs (line 189 RQ1, line 191 RQ2, etc.).Page 6, lines 206-207, "... filtering redundancy between items and integrating items with similar meanings." Please provide examples of these.
I would appreciate if you could provide a translated version of page 12, "Figure 2. Design layout of Pat Exp Insight Cards for patients with HNC". Discussion: Your discussion did not make a clear distinction between the importance and relevance of your results as compared to what you had done. As well, the discussion should focus on explaining and evaluating the findings, relating them to the literature review, and providing an argument in support of the conclusions you reached. Please modify!
Author Response
|
No. |
Reviewer’s comments |
Authors’ responses to the comments |
|
1 |
Page 2, lines 50-65 appear to be methods rather than an introduction. |
This part is not intended to explain the methodology of this study. Since this part introduces the methodological research trend that there is a growing interest in co-creation methods to improve patient experience and card tools are being usefully used, it seems appropriate to mention it in the introduction rather than the methods section. However, some cases of specific design methodologies mentioned unnecessarily were deleted to avoid blurring the context. (2page, line 50-66) |
|
2 |
Could you add citations and literature on page 2, lines 69-74? The research problem needs to be clarified with evidence and citations. |
We added two citations for line 69-74 as suggested. (2page, line 69-74)
[26] Cognetti, D. M.; Weber, R. S.; Lai, S. Y. Head and neck cancer: an evolving treatment paradigm. Cancer, 2008, 113(S7), 1911-1932. https://doi.org/10.1002/cncr.23654 [27] Rettig, E. M.; D’Souza, G. Epidemiology of head and neck cancer. Surgical Oncology Clinics, 2015, 24(3), 379-396. https://doi.org/10.1016/j.soc.2015.03.001 |
|
3 |
Page 2-3, lines 94 to 102 regarding research questions, I don't think this is necessary since you have already stated the purpose of the study. |
We deleted the research questions as suggested. (2page, line 94-102) |
|
4 |
As stated on page 3, line 111, "... four electronic databases," which databases? , what terms were used? You don't need to mention the databases in your results (line 148). |
The contents related to the method mentioned in the results are moved to the methods section, and the systematic review process was described in detail. (3page, line 111-123) |
|
5 |
Page 5, lines 166-175, these are not your results, so please focus on your results. I would move that to the methods section. |
The content has been moved to research methods. (3page, line 111-204) |
|
6 |
Instead of mentioning the numbers on page 6, please add the RQs (line 189 RQ1, line 191 RQ2, etc.). |
By accepting the opinions of other reviewers that this part corresponds to the research method, the core content was moved to the research method. (7page, line 256-274) |
|
7 |
Page 6, lines 206-207, "... filtering redundancy between items and integrating items with similar meanings." Please provide examples of these. |
Comparing phases 1 and 2 in Table 2, some examples of the changed contents are additionally explained in the text. (7page, line 281-292) |
|
8 |
I would appreciate if you could provide a translated version of page 12, "Figure 2. Design layout of Pat Exp Insight Cards for patients with HNC". |
The contents of Figure 2 were translated into English and the existing image was replaced. (16page, Figure 2) |
|
9 |
Your discussion did not make a clear distinction between the importance and relevance of your results as compared to what you had done. As well, the discussion should focus on explaining and evaluating the findings, relating them to the literature review, and providing an argument in support of the conclusions you reached. Please modify! |
We added a summary of our findings (24page, line 6-11), and comparing to previous studies, the value and distinguishment of the results of this study were additionally mentioned. (24page, line 33-40) |

Reviewer 2 Report
The article addresses the relevant topic of patient's involvement in medical decision-making. I have only few suggestions:
- the title is very long, consider to shorten it
- provide more information on the systematic review (follow the PRISMA checklist at this purpose);
- report whether ethical approval was obtained before conducting the study, and how patients and caregivers were invited to take part and involved in the study;
- HNC is a family of neoplasms (laryngeal, pharyngeal, oral, etc.). Did you consider any potential differences in terms of patient's needs? If not, provide a justification for this.
Reviewer 3 Report
This is an interesting paper that focuses on co-creation tool development for improving the healthcare experience of patients with head and neck cancer. This is a worthy paper also because the co-design of tools to improve patient experience is an emerging issue that deserves attention. However, the paper is very long, often repetitive in the content and with overlaps between methods and findings. The work may benefit from restructuring and condensation to improve the readability. Some suggestions follow:
Phase 1:
- Authors mentioned that they did a systematic review but no search strategies is presented in the appendix to prove the process.
- The main questions of the semi-structured interviews should be shown.
- What observation consisted of?
Results:
Extensive results sections sound more like methods and should be moved (for example, lines 145-156, 165-179, 184-203, 364-391, 426-429).
Table 1, table 3, and table 4 can be merged highlighting changes over the co-design process.
Table 5 is not very useful in the overall economy of the paper and can be moved to the appendix.
Section 3.4.2. Comparing viewpoints of patients and medical staff. What do numbers in brackets mean?
Round 2
Reviewer 1 Report
Dear authors,
The manuscript has improved significantly. However, a minor comment must be made before accepting the final version. Please review your responses and correction lines. As you mentioned, the line numbers of your correction are incorrect compared with the coloured text in your revised manuscript.
Author Response
Thank you for your review comments.
We have modified our reply to accommodate the corrections made based on your comments below.
“The manuscript has improved significantly. However, a minor comment must be made before accepting the final version. Please review your responses and correction lines. As you mentioned, the line numbers of your correction are incorrect compared with the coloured text in your revised manuscript.”
We hope this revision will answer your comments sufficiently and meet your review point.
Please see the attachment for details.
Thank you.

Reviewer 3 Report
The Authors have adeguately adressed all previous comments.
Just a minor suggestion to further improve the consistency. In table 2 phase 3, remove the column No. and numbers in the columns “category” and “sub-category”
Author Response
Unlike other phases, the numbers were assigned in the columns in phase 3 to clearly distinguish and explain each item in the text as the finally developed items. We explained the card list on pages 10-11, page 20 (lines 500-509) & page 22 (Table 3) based on the numbers shown in Table 2, phase 3. These numbers are essential in this table and phase for the description of each item of the card list to be conveyed to the reader.